# Video-Assisted Intubating Stylet Technique for Difficult Intubation: A Case Series Report

**DOI:** 10.3390/healthcare10040741

**Published:** 2022-04-15

**Authors:** Pei-Jiuan Tsay, Chih-Pin Yang, Hsiang-Ning Luk, Jason Zhensheng Qu, Alan Shikani

**Affiliations:** 1Department of Anesthesia, Hualien Tzuchi Hospital, Hualien 97002, Taiwan; juliatsay@tzuchi.com.tw (P.-J.T.); dingisfun903@tzuchi.com.tw (C.-P.Y.); 2Bio-Math Laboratory, Department of Financial Engineering, Providence University, Taichung 43301, Taiwan; 3Department of Anesthesia, Critical Care and Pain Medicine, Massachusetts General Hospital, Harvard Medical School, Boston, MA 02114, USA; JQU@mgh.harvard.edu; 4Division of Otolaryngology-Head and Neck Surgery, LifeBridge Sinai Hospital, Baltimore, MD 21215, USA; ashikani@marylandent.com; 5Division of Otolaryngology-Head and Neck Surgery, MedStar Union Memorial Hospital, Baltimore, MD 21218, USA

**Keywords:** difficult airway, intubating stylet, video laryngoscopy, craniofacial tumor, hypopharyngeal cancer, tonsillectomy, radiation fibrosis

## Abstract

Induction of anesthesia can be challenging for patients with difficult airways and head or neck tumors. Factors that could complicate airway management include poor dentition, limited mouth opening, restricted neck motility, narrowing of oral airway space, restricted laryngeal and pharyngeal space, and obstruction of glottic regions from the tumor. Current difficult airway management guidelines include awake tracheal intubation, anesthetized tracheal intubation, or combined awake and anesthetized intubation. Video laryngoscopy is often chosen over direct laryngoscopy in patients with difficult airways because of an improved laryngeal view, higher frequency of successful intubations, higher frequency of first-attempt intubation, and fewer intubation attempts. In this case series report, we describe the video-assisted intubating stylet technique in five patients with difficult airways. We believe that the intubating stylet is a feasible and safe airway technique for anesthetized tracheal intubation in patients with an anticipated difficult airway.

## 1. Introduction

A difficult airway is always challenging for the airway operator. The incidence of failed tracheal intubation varies from 0.1% to 11% depending on patients’ conditions and clinical scenarios [1,2]. Difficult airway guidelines have been developed and periodically revised to provide recommendations for clinicians. In clinical practice, video-laryngoscopes, fiberoptic bronchoscopes, supraglottic devices, or combined tools techniques are commonly used for elective/routine and emergency tracheal intubation [3,4,5].

A plethora of clinical studies have shown the advantages of commercial video-laryngoscopes over conventional direct laryngoscopes [6,7]. The comparisons were made in different clinical scenarios (e.g., elective and emergency intubation), distinct environments (e.g., operating rooms and intensive care units), various patients’ conditions (e.g., adult, pediatrics, pregnancy, anesthetized or awake), and airway status (normal airway, expected and unanticipated difficult airway). The comparison parameters include the time taken to intubate, the success rate of the first attempt, the overall success rate, alternative rescue methods, and the complications rate.

In the past decades, since the video-assisted rigid intubating stylet was designed by Shikani [8] and modified by Levitan [9], numerous articles have been published detailing the safety and advantages of the Shikani video-assisted optical stylet tracheal intubation method both in routine and unanticipated difficult airway management [10,11]. In this case series report, the objective was to share our experience using the Shikani video-assisted intubating stylet technique in five anticipated but challenging, difficult airway situations. There are four different brands of intubating stylets in our hands (Figure 1) and each of them suits its purpose perfectly. In this case series report, all the tracheal intubations were performed using the Video Intubation Light Stylet because of its unique feature of wireless transmission of the images onto a separate screen. Video Intubation Light Stylet (Trachway) is the same product as the Clarus Video System (CVS) and Optiscope Video Stylet (Clarus Medical LLC, Minneapolis, MN, USA) under different brand names. The airway operator was the same one for all these five cases, who is a certified anesthesiologist with 40 years of experience (including 16 years and 6 years of experience with video laryngoscopy and the intubating stylet technique, respectively). In an anesthetized human subject without anticipated airway problems, tracheal intubation using Video Intubation Light Stylet was shown to be smooth, swift, and accurate (Figure 2 and Appendix A).

## 2. Cases Presentation

### 2.1. Case 1: Giant Cemento-Ossifying Fibroma

A 26-year-old man (158 cm, 52 kg, body mass index [BMI] 20.8 kg/m^2^) was referred to our medical center for an oculofacial tumor for four years. The preoperative tumor biopsy indicated a diagnosis of cemento-ossifying fibroma. A combined operation with wide excision, free fibular flap, and vastus lateralis muscle flap was scheduled in 2021. Because the tumor was extensive (14 cm × 11 cm × 10 cm in size) and invaded the left eye, nose, paranasal sinus, maxilla, and oral cavity, the flexible fiberoptic intubation through the nasotracheal path was impossible (Figure 3A–C). An awake surgical tracheostomy was refused by the patient as the first choice. After evaluation of the airway and ventilation status, oral tracheal intubation with the Shikani video-assisted intubating stylet technique was performed. The surgeons stood by for the possibility of surgical airway intervention. The patient first received pre-medication and high-flow nasal cannula oxygenation, followed by routine induction agents for anesthesia (glycopyrrolate, lidocaine, fentanyl, propofol, and succinylcholine). Tracheal intubation was planned using the video-assisted intubating stylet technique (with the Video Intubation Light Stylet). Although the oral cavity space was not wide enough, the stylet still could reach the epiglottis and then slipped under it. Eventually, a clear and full-size glottis view was obtained and the endotracheal tube was smoothly and swiftly railroaded into the trachea (Figure 3D). The tracheal intubation was confirmed, secured and mechanical ventilation was initiated without complications. The whole intubation process is demonstrated in Appendix A.

### 2.2. Case 2: Enlarged Palatine Tonsils

A 12-year-old girl (158 cm, 95 kg, BMI 38.0 kg/m^2^) was diagnosed with obstructive sleep apnea syndrome (OSAS) caused by enlarged tonsils (grade 3) and a redundant uvula. She was scheduled to undergo uvulo-palato-pharyngoplasty (UPPP) under general anesthesia in 2021. Routine vital signs monitoring was implemented, and anesthesia was induced with fentanyl, lidocaine, propofol, and rocuronium. Oral tracheal intubation was performed using the Shikani video-assisted intubating stylet technique (with the Video Intubation Light Stylet) with a 6.5 mm diameter endotracheal tube. Due to copious saliva and secretion in her oropharyngeal cavity, a suction tube was applied to clear the airway before tracheal intubation. The epiglottis was reached, and a clear glottis view was obtained without difficulty (Figure 4). The whole process of tracheal intubation was smooth and swift (as shown in Appendix A).

### 2.3. Case 3: Lower Gum Carcinoma with Neck Radiation Fibrosis

A 68-year-old man (164 cm, 40 kg, BMI 14.8 kg/m^2^) with right lower gum squamous cell carcinoma (pT4aN0) was scheduled to undergo a wide tumor excision (Commando operation), mandibulectomy, suprahyoid neck dissection, and tracheostomy in 2021. He was previously diagnosed with buccal and soft palate cancer 20 years ago and had undergone surgical resection and radiotherapy. Although post-radiotherapy neck muscle atrophy and soft-tissue fibrosis were noticed, the patient did not demonstrate any stridor or difficult breathing before induction of anesthesia. The mouth opening was restricted and only two-finger wide (Figure 5A,B). Although awake tracheostomy was not impossible, the surgical team preferred tracheal intubation to secure the airway first. We felt that a flexible fiberoptic intubation technique via the nasal path would be an option, however, we anticipated difficulty with increased secretions and bleeding along with the distorted anatomy with potential difficulty in advancing the tracheal tube. Oral tracheal intubation with the Shikani video-assisted intubating stylet technique was chosen (with the Video Intubation Light Stylet). Anesthesia was induced with fentanyl, lidocaine, propofol, and rocuronium. Sugammadex was ready for rescue if any difficult airway was encountered. The intubating procedure ended up being smooth and swift (Figure 5C,D also shown in Appendix A).

### 2.4. Case 4: Hypopharyngeal Cancer

A 63-year-old man (169 cm, 62 kg, BMI 21.7 kg/m^2^) was in his usual health status until 2 months prior to surgery when hoarseness was noted. A further examination and biopsy showed stage IV hypopharyngeal squamous cell carcinoma involving the left pyriform sinus cancer (T4aN2b) with extranodal extension (Figure 6A–C). The patient was scheduled to undergo a total laryngectomy with type II modified radical neck dissection, right selective neck dissection (II, III, IV), and free flap reconstruction in 2021. The standard monitoring was implemented and preoxygenation was applied. Oro-tracheal intubation was planned using the Shikani video-assisted intubating stylet technique (with the Video Intubation Light Stylet) and was done with a 7.5 mm diameter endotracheal tube (Figure 6D,E). The surgeons were on standby for a possible surgical tracheotomy, however, this was not needed. Anesthesia was induced with glycopyrrolate, fentanyl, lidocaine, propofol, and succinylcholine. A suction tube through a nasal airway was inserted to clear the secretion inside the oropharyngeal space. The intubating process was done via the Shikani video-assisted stylet intubating technique in a smooth, swift, and uneventful fashion (shown in Appendix A).

### 2.5. Case 5: Vocal Cord Granuloma

A 46-year-old woman (160 cm and 72 kg, BMI 28.1 kg/m^2^) received a craniectomy due to intracerebral hemorrhage under general anesthesia with tracheal intubation with mechanical ventilation during the course of treatment. One year later, she suffered from hoarseness and was diagnosed with a very large vocal cord granuloma, obstructing a good portion of the glottis. After making her own decision with some delay, she eventually was willing to receive laryngo-microsurgery (LMS) to remove the granuloma in 2019. After implementing standard monitoring and adequate preoxygenation, anesthesia was induced with routine medications, including glycopyrrolate, lidocaine, fentanyl, propofol, and succinylcholine. The ENT team was on standby for possible surgical airway management, however, this was not needed. Oral tracheal intubation was then performed with the Shikani video-assisted intubating technique (with the Video Intubation Light Stylet) and a 6.5 mm endotracheal tube. The tracheal intubation was swift and successful on the first attempt (Figure 7A,B, also shown in Appendix A). Resection of the granuloma by LMS was smooth and uneventful (Figure 7C,D).

## 3. Discussion

We reported five cases of anticipated difficult airways with various presentations, from facial-oral lesions to glottis abnormalities. Among the recommendations and the update in difficult airway guidelines made by the American Society of Anesthesiologists, some strategies have been proposed for the anticipated difficult intubation, depending on the patient’s factors and condition, and the anesthesiologist’s experience, skills, and preferences regarding airway management [12,13]. For example, in a situation where a difficult airway is anticipated but the patient can be adequately ventilated and oxygenated, awake intubation may not always be the first option. Instead, intubation under general anesthesia may be safer. In this case, the video laryngoscopy has been reported to be successful in various difficult airway scenarios (including morbid obesity, ankylosing spondylitis, congenital facial maxillary anomalies, giant thyroid goiter, oropharyngeal or laryngeal mass or tumor) [14,15].

In this report, we demonstrated the role of the Shikani video-assisted intubating stylet technique (Figure 1 and Figure 2) in five patients with different airway challenges (from a facial-oral tumor to a glottis lesion, Figure 3, Figure 4, Figure 5, Figure 6 and Figure 7). Although this case series report was not designed as a comparative clinical study between video laryngoscopy and the video-assisted intubating stylet technique, it is important to mention that we were successful on the first attempt using the aforementioned video-assisted intubating stylet in all these patients while we had real concerns about potentially facing difficulty with the other technique. In addition to the high success rate of the first attempt, the advantages of the intubating stylet technique include the short time to complete intubation, a low incidence of soft-tissue injuries and sympathetic reflexion (Table 1). Above all, the satisfaction level was high among the operators using the video-assisted intubating stylet technique.

The glottic view quality was judged by either the Cormack–Lehane grades or POGO scores.

The degree of difficulty for intubation was evaluated subjectively and confirmed by reviewing the video-recording images (Appendix A).

Post-operative sore throat (POST) and hoarseness were not evaluated in these five head-neck surgical cases.

The potential airway risks for patients undergoing oral and throat surgery (e.g., tonsillectomy, laryngeal surgery, thyroidectomy, laryngectomy, etc.) cannot be overemphasized [16,17]. Unanticipated airway problems have been reported in patients with enlarged tonsils [18], thyroid tumor [19], neck radiation fibrosis [20], hypopharyngeal cancer and laryngectomy [21], and glottic lesions [22,23]. When dealing with a potentially difficult airway, airway operators should be prepared for any emergency. They should be familiar and comfortable with various airway management techniques that can be considered, including conventional blade laryngoscopy using a blade [24], fiberoptic nasotracheal intubation [25,26], flexible fiberoptic bronchoscope [27,28] and video laryngoscopy [29,30].

If the operator is not sure about the airway, it is generally recommended to keep the patient awake and maintain spontaneous respiration. In those cases, blade laryngoscopy intubation, which requires general anesthesia, is best avoided. Fiberoptic nasotracheal intubation and intubating stylet video laryngoscopy may be safer options in difficult airways. If the airway is potentially unstable, a planned surgical airway (tracheotomy) with the surgeon on standby could be considered. It has been our experience though, that intubation using the Shikani video-assisted intubating stylet technique has allowed us to avoid emergency tracheotomies in the great majority of difficult airway cases that we have encountered, even critical ones. This technique has shown to be superior to the Macintosh blade intubation in the literature [31,32,33,34,35,36,37,38].

In this case series, we report that the Shikani video-assisted intubating stylet technique is an easy and safe way to secure a variety of airway pathologies, including enlarged tonsils, radiation fibrosis of the upper airway and neck, hypopharyngeal cancer, vocal polyps, and granulomas. This is the first time that the use of the video-assisted intubating stylet technique has been reported in maxillofacial surgery. Difficult airways using conventional laryngoscopy were reported in maxillofacial surgery with a prevalence rate of 15.4% (against 1% to 4% in overall incidence) [13,14,39]. We have found that the video-intubating stylet technique is smooth and successful for patients undergoing large maxillofacial surgery (Figure 3, Table 1). It is also relevant to mention that the Shikani intubating stylet technique has also been described as advantageous in restricted neck movement and limited mouth opening [31,32,33,34,35,36,37,38,39,40]. It is less traumatic, being able to be used in patients with small mouth openings, thus requiring fewer neck or laryngeal manipulations [41,42,43,44].

Furthermore, it is worth mentioning that the airway operator might encounter some troubles while using the intubating stylet technique, such as an obstructed view by the soft tissues, difficulty finding the correct path to insert, blockade by the epiglottis, fogging of the lens, obstruction of the optics by saliva, mucous plugs, or blood, difficulty railroading the endotracheal tube into the trachea, etc. Fortunately, according to our own experiences, the time course for developing competency/proficiency in intubating using the stylet technique is comparable to video laryngoscopy. Moreover, it should be emphasized that the outcomes of using the intubating stylet technique in anticipated difficult airway scenarios depend on various factors and conditions. While some airway operators with vast use experience and skills found the intubating stylet technique to be easy, others might not feel so at ease with dealing with such a technique. Therefore, a Plan B for airway management in such patient populations with potentially difficult airways should always be prepared in advance.

We found the Shikani video-assisted intubating stylet technique to be safe and reliable in a variety of airway management cases. We recommend that the airway operator add this technique to their armamentarium for managing the difficult airway. Initial training with simple cases is recommended before handling difficult ones.

## Figures and Tables

**Figure 1 healthcare-10-00741-f001:**
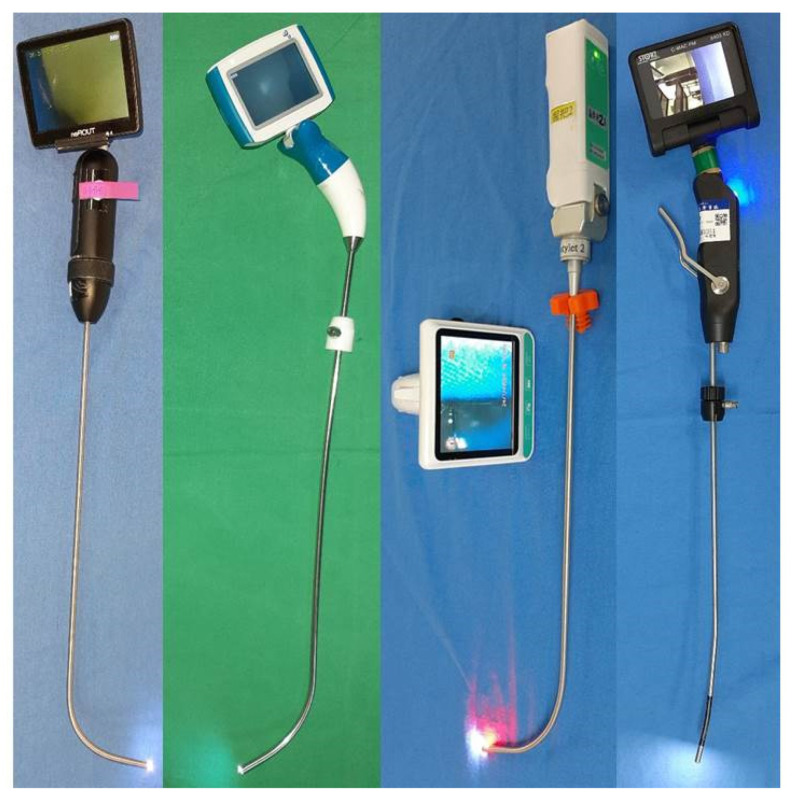
The video-assisted intubating stylets we routinely use for tracheal intubation. From left to right: Tuoren (Tuoren Kingtaek Video Intubating stylet, Henan Tuoren Medical Device Co., Changyuan City, Xinxiang, Henan, China); UE TRS video stylet (Zhejiang UE Medical Corp., Taizhou, Zhejiang, China); Video Intubation Light Stylet (Markstein Sichtec Medical Corp., Taichung, Taiwan); C-MAC^®^ VS (Video Stylet) (KARL STORZ SE & Co. KG, Tuttlingen, Germany).

**Figure 2 healthcare-10-00741-f002:**
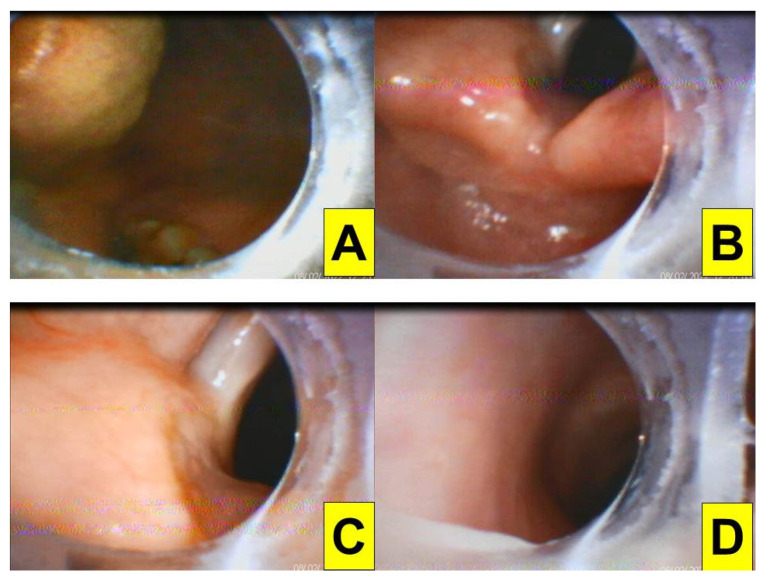
Views obtained through the video-assisted intubating stylet technique. (**A**) Oral pharynx; (**B**) glottis; (**C**,**D**) tracheal lumens. Time to intubate (from lip to trachea) is 7 s (see also the video clip in the Appendix A).

**Figure 3 healthcare-10-00741-f003:**
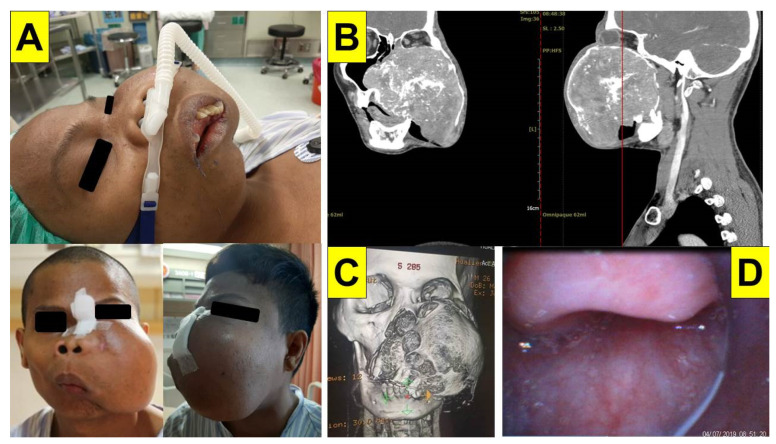
Case 1 (cemento-ossifying fibroma). (**A**) preoperative facial photos of the fibroma from different angles; (**B**) CT scan images of the fibroma; (**C**) preoperative computerized tomography face three-dimensional reconstruction of frontal view; (**D**) view of epiglottis from intubating stylet camera. (See also the video clip in Appendix A).

**Figure 4 healthcare-10-00741-f004:**
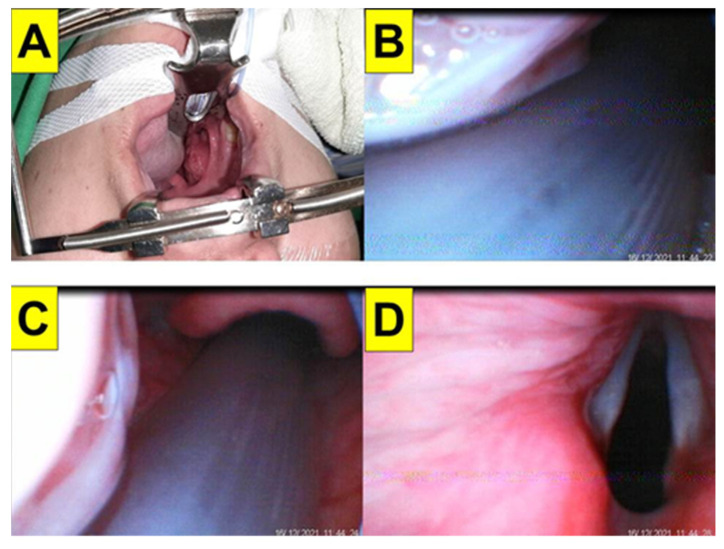
Case 2 (enlarged palatine tonsils). (**A**) View of enlarged right tonsil through a Dingman mouth gag; (**B**) intubating stylet passing by the left tonsil; (**C**) view of tonsil and epiglottis; (**D**) complete glottic view from intubating stylet camera. A suction tube device is seen in (**B**,**C**). (See also the video clip, Appendix A).

**Figure 5 healthcare-10-00741-f005:**
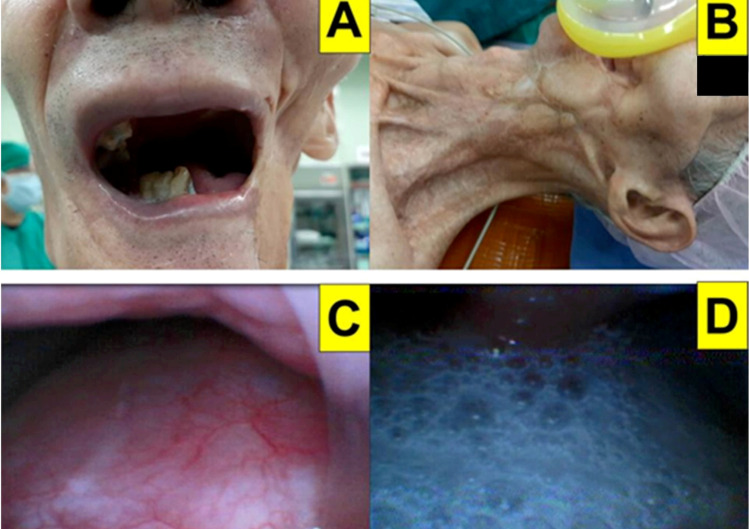
Case 3 (lower gum carcinoma with neck radiation fibrosis and surgical scar contracture). Limited mouth opening and poor dentition (**A**); stiff neck with restricted motility due to radiation fibrosis and scar contracture (**B**); intubating stylet view on pharyngeal space (**C**) and larynx (**D**). Copious mucus and saliva were noticed in front of the glottis. (See also the video clip in the Appendix A).

**Figure 6 healthcare-10-00741-f006:**
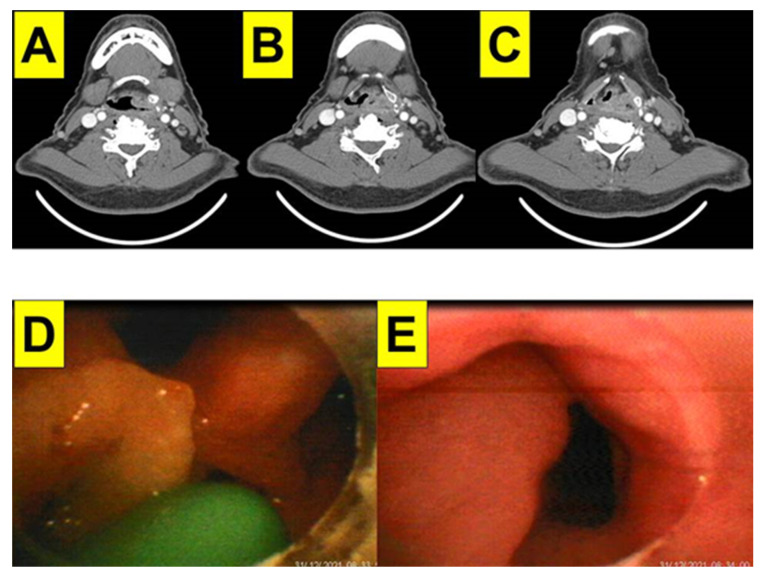
Case 4 (hypopharyngeal cancer). Computerized tomography scan views on hypopharyngeal tumor lesion (**A**–**C**). Such lesions can be seen from the intubating stylet camera (**D**,**E**). (See also the video clip in Appendix A).

**Figure 7 healthcare-10-00741-f007:**
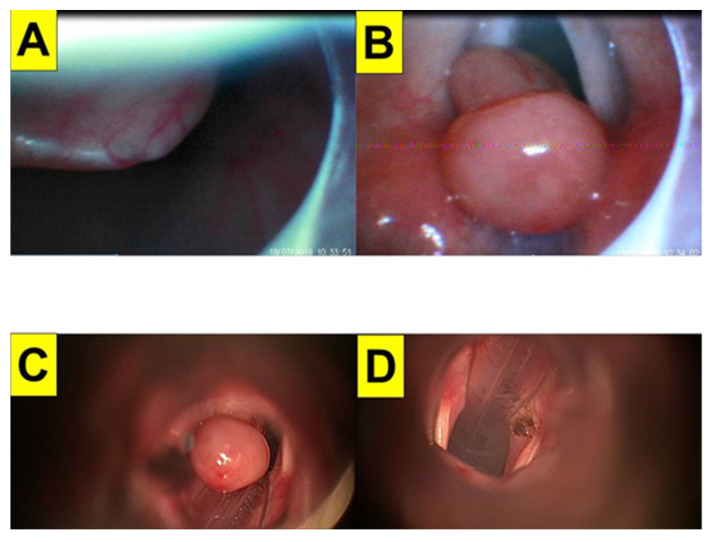
Case 5 (vocal cord granuloma). Intubating stylet view on epiglottis (**A**) and glottis (**B**). A granuloma was seen; laryngeal microscopic view on the granuloma before (**C**) and after (**D**) resection. (The video clip for the intubation process is shown in Appendix A).

**Table 1 healthcare-10-00741-t001:** Advantageous features of video-assisted intubating stylet technique in the presenting five different clinical cases.

	Case 1	Case 2	Case 3	Case 4	Case 5
First-attempt success	Yes	Yes	Yes	Yes	Yes
Overall success rate	100%				
Laryngeal & glottic view	Full	Full	Full	Full	Full
Time to intubate (from lip to trachea)	24 s	12 s	25 s	26 s	50 s
Degree of difficulty for intubation	Easy	Easy	Easy	Easy	Easy
Required additional external maneuvers	No	No	No	No	No
Complications (tissue injuries, autonomic overstimulation, hypoxia)	No	No	No	No	No

## Data Availability

Not applicable.

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
