# Peer review of "Video-Assisted Intubating Stylet Technique for Difficult Intubation: A Case Series Report"

_healthcare, 2022, doi:10.3390/healthcare10040741_

Round 1

Reviewer 1 Report

The authors report the intubation of five patients (with very different clinical characteristics) using the Shikani video-assisted intubating stylet.
The manuscript is very interesting, well written, and well referenced. The videos given as supplementary material are excellent. A rather light editing should be able to address the few mistakes and typos still present in the manuscript.

Major comments:
1. The general context and the selection of the cases should be detailed: approximate dates (month/year, or at least year) and operator characteristics should be added (for each case). Was it always the same operator? What was their experience? How many ETIs had they performed using this stylet? How had they been trained?
2. Accordingly, the limitations of this study need to be added to the discussion: if the operator was always highly experienced in using such stylets (which is rather probable according to the videos), it should be acknowledged that less-experienced operators might not find ETI so easy to perform even when using such stylets. In addition, the fact that this is only a case series should be emphasized in the limitations section.
3. According to Figure 1, the authors routinely use many different optical stylets, but I did not see the original Shikani (Clarus Medical) even though Alan Shikani is the last author. While I can only commend Alan Shikani on his humility, I believe that Figure 1 should be discarded if the original Shikani stylet was the only one used to intubate these 5 cases. If different stylets were used to intubate these cases, the exact stylet should be referenced in each case (along with their manufacturer).
4. Were there any post-intubation complications other than those reported in Table 1 (such as the rather usual occurences of sore throat, dysphagia, etc?).
5. The two first rows of Table 1 should be either removed or modified: A success rate (overall or at first attempt) can only be either 0% or 100% for a specific patient, and giving percentages rather than "Success" / "Failure" does not make much sense.
6. Still in Table 1, how and by whom was the "degree of difficult intubation" rated?

Minor comments:
7. The sentence "In the past decades, clinical studies have shown that video-assisted rigid intubating stylet [10], designed by Shikani [11] and modified by Levitan [12], and numerous articles have been published detailing the safety and advantages of the Shikani video-assisted optical stylet tracheal intubation method both in routine and unanticipated difficult air-way management [13-17]." is too long , lacks a verb, and should therefore be split and completed.
8. The article should be written using either the first or the third person consistently. The sentence "In this case series report, the authors share their experience us-ing the Shikani video-assisted intubating stylet technique in five anticipated but challeng-ing difficult airway situations." is inconsistent with Figure 1's legend ("The video-assisted intubating stylets we use routinely"). I would advise keeping the legend as is and to modify the objective thus: "Our objective was to share our experience using the Shikani [...]" (any similar sentence would of course be perfectly acceptable).
9. In the text, Figure 3 is referenced before Figures 1&2. The manuscript should be modified either to reference Figures 1 & 2 more early (I would not choose this option) or by moving these Figures below Figure 3 (the Figures would of course need to be renumbered accordingly).
10. In line with the previous comment, the videos provided as supplementary material should be referenced within the core text to facilitate their identification.
11. Units should be given when reporting body mass index values (kg/m2)

Author Response

Ms No: 1677578

Ms Title: Video-assisted intubating stylet technique for difficult intubation: A case series report.

Ms Authors: Tsay et al.

April 12, 2022

Response to the Reviewer-1:

The authors report the intubation of five patients (with very different clinical characteristics) using the Shikani video-assisted intubating stylet. The manuscript is very interesting, well written, and well referenced. The videos given as supplementary material are excellent. A rather light editing should be able to address the few mistakes and typos still present in the manuscript.

Response: We wish to acknowledge the reviewer for your excellent comments and professional opinions on our manuscript. With your constructive inputs for the review, we believe the value and the contribution of this scientific communication can therefore be enhanced and finally accomplished.

Major comments:

Comment-1: The general context and the selection of the cases should be detailed: approximate dates (month/year, or at least year) and operator characteristics should be added (for each case). Was it always the same operator? What was their experience? How many ETIs had they performed using this stylet? How had they been trained?

Response 1: Appreciated for your crucial questions and detailed comments. We respond as follows:

  • Approximate dates of the case: Due to the local regulations and hospital policies, unfortunately we are not allowed to indicate the operation date (and other PHI) of the cases in the article. This is to protect the patients’ confidentiality and privacy.
  • Operator characteristics for each case: The operator for all these five cases presented in this paper is the same airway operator (Dr. H.N. Luk). This particular airway operator has 40 years of anesthesia practice experience, including 16 years for video-laryngoscopy and 6 years for video intubating stylet technique.
  • Hualien Tzu-Chi Medical Center is the first Taiwanese tertiary medical center overwhelmingly using intubating stylet technique as the predominant tool to intubate patients. This hospital has 987 beds and more than 21000 surgical operations annually. The Department of Anesthesia has 14 attending physicians, 8 residents, and 54 CRNA. The video-assisted intubating stylet technique is accountable for more than 90% of the tracheal intubation for general anesthesia (examples shown in Figures 1 and 2). It is estimated that more than 8000 tracheal intubations were done by intubating stylet technique every year in our hospital. Namely, the intubating stylet technique is not only an option for emergency difficult airway management, but also for routine elective tracheal intubation. A small proportion of the tracheal intubations were conducted by conventional direct laryngoscopy and video-laryngoscopy (only for teaching purposes and some personal preferences).
  • For novice practitioners (e.g., clerks, PGY-1, PGY-2, and residents), we provided stepwise training courses for airway management in Hualien Tzu-Chi Medical Center. The courses include cadaveric practices in Medical Simulation Center (Department of Anatomy), simulator practices in Clinical Skill Center (Department of Medical Education), and clinical observation and bedside teaching in Department of Anesthesia. Therefore, each student and resident has at least 3 times of training opportunities before their hands-on patients. The airway management courses include all the classical teaching materials, from mask ventilation to eFONA. In particular, we enhanced the teaching and practice of using video-assisted intubating stylet technique. It is worthy to mention that even the EMT personnel now are receiving the intubating stylet training and well equipped with such tool in their units in Taiwan.
  • For our attending physicians, if they are not familiar with the intubating stylet technique, we assisted them to get into the swing of things. All the anesthesiologists are quick learners for intubating stylet technique! For this moment, our department has been equipped with 19 sets of Trachway, 2 sets of UEScope, 2 sets of TuoRen Stylet, and 2 sets of Storz C-MAC VS. We are also equipped with video-laryngoscopes, e.g., Glidescope, McGrath, UEScope, AWS airwayscope, and TuoRenScope.

The following statement has been added in the Introduction section:

There are four different brands of intubating stylets in our hands (Figure 1) and each of them suits its purpose perfectly. In this case series report, all the tracheal intubations were performed using the Video Intubation Light Stylet because of its unique feature of wireless transmission of the images onto a separate screen. Video Intubation Light Stylet (Trachway) is the same product as the Clarus Video System (CVS) and Optiscope Video Stylet (Clarus Medical LLC, Minneapolis, MN, USA) under different brand names. The airway operator was the same one for all these five cases, who is certified anesthesiologist with 40 years of experience (including 16 years and 6 years of experience with videolaryngoscopy and intubating stylet technique, respectively). In an anesthetized human subject without anticipated airway problem, tracheal intubation using Video Intubation Light Stylet was shown to be smooth, swift, and accurate (Figure 2 and the supplementary material-1).

Comment-2: Accordingly, the limitations of this study need to be added to the discussion: if the operator was always highly experienced in using such stylets (which is rather probable according to the videos), it should be acknowledged that less-experienced operators might not find ETI so easy to perform even when using such stylets. In addition, the fact that this is only a case series should be emphasized in the limitations section.

Response 2: Thanks for bringing us this important issue. We totally agree with you and such notion has been added in the Discussion section, as follows:

…………. “Also, it should be emphasized that the outcomes of using intubating stylet technique in anticipated difficult airway scenarios depend on various factors and conditions. While some operators with vast use experiences and skills found intubating stylet technique to be easy, the others might feel not so easy to deal with such technique. Therefore, plan B for airway management in such patient populations with potential difficult airway should always be prepared in advance.

Comment-3: According to Figure 1, the authors routinely use many different optical stylets, but I did not see the original Shikani (Clarus Medical) even though Alan Shikani is the last author. While I can only commend Alan Shikani on his humility, I believe that Figure 1 should be discarded if the original Shikani stylet was the only one used to intubate these 5 cases. If different stylets were used to intubate these cases, the exact stylet should be referenced in each case (along with their manufacturer).

Response 3: Thank you for your important point. Indeed, Dr. Shikani is a very humble scholar and is one of the pioneers in the field of intubating stylet technique. We feel honored to co-author with him.

The reason we presented figures 1 and 2 is to introduce the intubating stylets to the readers who might not be familiar with this technique. Before letting the readers go into the 5 cases, we believe, it will be helpful to illustrate what armamentarium we had (Figure 1) and what the real scenario looked like with this technique (Figure 2). For the readers who had no experience in using intubating stylet, they might be confused with different terms, e.g., Shikani Optical Stylet (SOS, Clarus Medical), Levitan First Pass Success (FPS, Clarus Medical), Storz C-MAC VS (Storz), UEscope, TuoRen stylet (Tuoren Kingtaek Video Intubating stylet, Henan Tuoren Medical Device Co., China), etc. Actually, the Clarus Video System (CVS, Clarus, marketed in USA) is the same product as the Trachway (marketed in Taiwan) and Optiscope video stylet (Optiscope, marketed in Korea; Clarus Medical LLC, Minneapolis, MN, USA) under different brand names. That is why we felt that it would be easier to illustrate several different products of intubating stylet for the readers before their jumping into our presentation.

Comment-4: Were there any post-intubation complications other than those reported in Table 1 (such as the rather usual occurrence of sore throat, dysphagia, etc?).

Response 4: Thank you for your question about the related complications in our cases. Fortunately, there were no significant post-intubation complications in our 5 cases, e.g., soft tissue injuries, autonomic over-stimulation, etc. Post-operative sore throat (POST) and hoarseness were not analyzed in this case series report because the head-neck surgical intervention itself definitely would compromise the validity of clinical evaluation of the POST and hoarseness.  

Comment-5:. The two first rows of Table 1 should be either removed or modified: A success rate (overall or at first attempt) can only be either 0% or 100% for a specific patient, and giving percentages rather than "Success" / "Failure" does not make much sense.

Response 5: Thank you for your correction. We have revised the Table 1 accordingly.

Comment-6: Still in Table 1, how and by whom was the "degree of difficult intubation" rated?

Response 6: Dr. H.N. Luk is the only one person who rated the “degree of difficult intubation”. It is true that the degree of difficult intubation might be difficult to define and most of time could be very subjective. People do commonly use certain parameters to define the degree of difficulty (e.g., time to intubate, time to laryngoscopy, time to visualize the structures, time to pass the tube into trachea, effort and force to lift the tongue and vallecular, number of attempt, required assistance, external manipulations, alternatives required, struggle to create the enough space for the blade, copious mucus and secretin, soiled, bloody and fogging of the view, etc). However, we believe the intubating stylet technique is much easier than conventional direct laryngoscopy and video-laryngoscopy. With the video-recording images, one can easily determine whether the intubating process is smooth or easy. Thanks for your comments.

Minor comments:

Comment-7: The sentence "In the past decades, clinical studies have shown that video-assisted rigid intubating stylet [10], designed by Shikani [11] and modified by Levitan [12], and numerous articles have been published detailing the safety and advantages of the Shikani video-assisted optical stylet tracheal intubation method both in routine and unanticipated difficult air-way management [13-17]." is too long , lacks a verb, and should therefore be split and completed.

Response 7: Thanks for your correction on the syntax error. The text has been revised accordingly as follows:  

"In the past decades, since the video-assisted rigid intubating stylet was designed by Shikani [8] and modified by Levitan [9], numerous articles have been published detailing the safety and advantages of the Shikani video-assisted optical stylet tracheal intubation method both in routine and unanticipated difficult airway management [10,11]."

Comment-8. The article should be written using either the first or the third person consistently. The sentence "In this case series report, the authors share their experience using the Shikani video-assisted intubating stylet technique in five anticipated but challenging difficult airway situations." is inconsistent with Figure 1's legend ("The video-assisted intubating stylets we use routinely"). I would advise keeping the legend as is and to modify the objective thus: "Our objective was to share our experience using the Shikani [...]" (any similar sentence would of course be perfectly acceptable).

Response-8: Thank you very much for your correction. The text has been revised accordingly (as follows).

“In this case series report, the objective was to share our experience using the Shikani video-assisted intubating stylet technique in five anticipated but challenging difficult airway situations.”

Comment-9. In the text, Figure 3 is referenced before Figures 1&2. The manuscript should be modified either to reference Figures 1 & 2 more early (I would not choose this option) or by moving these Figures below Figure 3 (the Figures would of course need to be renumbered accordingly).

Response-9: Thank you very much for your correction. The text has been revised (as follows) and therefore the references to Figure 1 and Figure 2 appeared earlier (in the section of Introduction).

There are four different brands of intubating stylets in our hands (Figure 1) and each of them suits its purpose perfectly. In this case series report, all the tracheal intubations were performed using the Video Intubation Light Stylet because of its unique feature of wireless transmission of the images onto a separate screen. Video Intubation Light Stylet (Trachway) is the same product as the Clarus Video System (CVS) and Optiscope Video Stylet (Clarus Medical LLC, Minneapolis, MN, USA) under different brand names. The airway operator was the same one for all these five cases, who is certified anesthesiologist with 40 years of experience (including 16 years and 6 years of experience with videolaryngoscopy and intubating stylet technique, respectively). In an anesthetized human subject without anticipated airway problem, tracheal intubation using Video Intubation Light Stylet was shown to be smooth, swift, and accurate (Figure 2 and the supplementary material-1).” 

Comment-10: In line with the previous comment, the videos provided as supplementary material should be referenced within the core text to facilitate their identification.

Response-10: All the references of the “supplementary materials for video clips” have been added in the revised text. Thank you. 

Comment-11: Units should be given when reporting body mass index values (kg/m2)

Response-11: The unit of the BMI (i.e., kg/m2) has been added accordingly. Thank you.

Reviewer 2 Report

Thank you for the opportunity to review this interesting case series on video-assisted intubating stylet technique for difficult intubation. The figures and videos are quite impressive and reflect the potential of this important airway management technique.

I have only a few comments:

  • the authors did not mention the training levels of the anesthetists who performed the intubations (years of professional experience, residents or board certified specialists).
  • some more information regarding the learning curve to achieve reliable success rates with this technique would be helpful.
  • please add limitations of this technique (fogging of the lens, blood and mucus, hypersalivation, etc.).
  • please remove at least 50% of the references. This is not a review article.
  • please remove the smileys from the patients´eyes and replace tham with black bars.
  • please remove double "and" in the first sentence of the abstract.

I would be happy to review a revised version - good luck and best regards...

Author Response

Ms No: 1677578

Ms Title: Video-assisted intubating stylet technique for difficult intubation: A case series report

Ms Authors: Tsay et al.

April 12, 2022

Response to the Reviewer-2:

Thank you for the opportunity to review this interesting case series on video-assisted intubating stylet technique for difficult intubation. The figures and videos are quite impressive and reflect the potential of this important airway management technique.

Response: We wish to acknowledge the reviewer for your excellent comments and professional opinions on our manuscript. With your constructive inputs for the review, we believe the value and the contribution of this scientific communication can therefore be enhanced and finally accomplished.

Comment-1: I have only a few comments. The authors did not mention the training levels of the anesthetists who performed the intubations (years of professional experience, residents or board certified specialists).

Response-1: In this case series report, there was only one airway operator involved (Dr. H.N. Luk). Dr. Luk is a board certified anesthesiologist with 40 years of experience. However, his experience on video-laryngoscopy and intubating stylet technique is only 16 years and 6 years, respectively. This background information has been added in the revised text as follows.

The airway operator was the same one for all these five cases, who is certified anesthesiologist with 40 years of experience (including 16 years and 6 years of experience with videolaryngoscopy and intubating stylet technique, respectively).

Comment-2: Some more information regarding the learning curve to achieve reliable success rates with this technique would be helpful.

Response-2: Thank you for your suggestion. Indeed, the learning curve for the intubating stylet technique (and perhaps also for laryngoscopy) is always an interesting and important issue to deal with. However, it is not easy to study the learning curve of various intubation techniques by the novice intubators because there are many factors involved, e.g., ethics issue of the study design, mannequin model or in real patients, simulated difficult airway or real anticipated airway scenarios, various difficult airway models, etc. The studies related to learning curve for intubating stylets technique are even less. It seems that, a general impression, intubating stylet technique is easier and faster to learn in comparison to video-laryngoscopy (or at least, comparable to video-laryngoscopy). We have added such context into the Discussion section, but have to leave out all the following literature because there are already too many in the References.

  1. Evans et al. A comparison of the Seeing Optical Stylet and the gum elastic bougie in simulated difficult tracheal intubation: a manikin study. Anaesthesia. 2006, 61, 478-481.
  2. Kim et al. Comparison of tracheal intubation with the Airway Scope or Clarus Video System in patients with cervical collars,” Anaesthesia. 2011, 66, 694–698.
  3. Tseng et al. A comparison of Trachway intubating stylet and Airway Scope for tracheal intubation by novice operators: a manikin study. Kaohsiung J Med Sci. 2012, 28, 448-451.
  4. Hung et al. A comparison of the Trachway intubating stylet and the Macintosh laryngoscope in tracheal intubation: a manikin study. Journal of Anesthesia. 2013, 27, 205–210.
  5. Moon et al. Endotracheal intubation by inexperienced trainees using the Clarus Video System: learning curve and orodental trauma perspectives. Journal of Dental Anesthesia and Pain Medicine. 2015, 15, 207–212.
  6. Ong et al. Comparison between the Trachway video intubating stylet and Macintosh laryngoscope in four simulated difficult tracheal intubations: A manikin study. Ci Ji Yi Xue Za Zhi. 2016, 28, 109-112.
  7. Altun et al. Learning curves for two fiberscopes in simulated difficult airway scenario with cumulative sum method. Simul Healthc. 2019, 14, 163-168.
  8. Theiler et al. The skill of tracheal intubation with rigid scopes - a randomised controlled trial comparing learning curves in 740 intubations. BMC Anesthesiol. 2020, 20(1):263. doi: 10.1186/s12871-020-01181-w.
  9. Pius et al. Learning curve and performance in simulated difficult airway for the novel C-MAC® video-stylet and C-MAC® Macintosh video laryngoscope: A prospective randomized manikin trial. PLoS One. 2020, 15(11):e0242154. doi: 10.1371/journal.pone.0242154.
  10. Park et al. Comparison of a new video intubation stylet and McGrath® MAC video laryngoscope for intubation in an airway manikin with normal airway and cervical spine immobilization scenarios by novice personnel: A randomized crossover study. Biomed Res Int. 2021, 2021:4288367. doi: 10.1155/2021/4288367.
  11. Chen et al. Retromolar intubation with video intubating stylet in difficult airway: A randomized crossover manikin study. Am J Emerg Med. 2022, 54, 212-220.

The following paragraph has been added in the Discussion section:

It is worthy to mention that the airway operator might encounter some troubles while using intubating stylet technique, such as obstructed view by the soft tissues, difficulty finding the correct path to insert, blockade by epiglottis, fogging of the lens, obstruction of the optics by saliva, mucous plugs, or blood, difficulty railroading the endotracheal tube into trachea, etc. Fortunately, according to our own experiences, the time course for developing competency/proficiency in intubating stylet technique is comparable to videolaryngoscopy.

Comment-3: please add limitations of this technique (fogging of the lens, blood and mucus, hypersalivation, etc.).

Response-3: We have added the “limitations” of the intubating stylet technique in the revised text as follows. Thank you.

It is worthy to mention that the airway operator might encounter some troubles while using intubating stylet technique, such as obstructed view by the soft tissues, difficulty finding the correct path to insert, blockade by epiglottis, fogging of the lens, obstruction of the optics by saliva, mucous plugs, or blood, difficulty railroading the endotracheal tube into trachea, etc. Fortunately, according to our own experiences, the time course for developing competency/proficiency in intubating stylet technique is comparable to videolaryngoscopy.

Comment-4: Please remove at least 50% of the references. This is not a review article.

Response-4: The references have been extensively reduced, from 71 down to 44. Thank you for your suggestion.

Comment-5: Please remove the smileys from the patients´eyes and replace them with black bars.

Response-5: Smileys in the Figure 3 have been removed and replaced by black bars. Thanks for your correction.

Comment-6: please remove double "and" in the first sentence of the abstract.

Response-6: The duplicate word of “and” has been deleted. Thanks!

Comment-7: I would be happy to review a revised version - good luck and best regards...

Response-7: We sincerely appreciate your efforts in improving the quality of our manuscript.

Round 2

Reviewer 1 Report

I congratulate the authors on this excellent revision of their manuscript and thank them for providing us with their thoroough responses.

Reviewer 2 Report

I would like to thank the authors for their modifications. I have no further comments and recommend publication of this interesting case series. Good luck and thank you again. Best regards, Manuel Struck